# Cyclophosphamide addition to pomalidomide/dexamethasone is not necessarily associated with universal benefits in RRMM

Hyunkyung Park[1‡], Ja Min Byun[2,3‡], Sung-Soo Yoon[2,3,4‡]*, Youngil Koh[2,3‡]*, Sock-Won Yoon[2], Dong-Yeop Shin[2,3], Junshik Hong[2,3,4], Inho Kim[2]

1 Department of Internal Medicine, Seoul National University Boramae Medical Center, Seoul, Korea, 2 Department of Internal Medicine, Seoul National University College of Medicine, Seoul National University Hospital, Seoul, Korea, 3 Cancer Research Institute, Seoul National University College of Medicine, Seoul, Korea, 4 Center for Medical Innovation, Biomedical Research Institute, Seoul National University Hospital, Seoul, Korea

‡ HP and JMB are contributed equally to this study as the first authors. SSY and YK are contributed equally to this study as corresponding authors.
* ssysmc@snu.ac.kr (SSY); go01@snu.ac.kr (YK)

**Data Availability Statement:** All relevant data are within the manuscript and its Supporting Information files.

**Funding:** This study was supported by a grant of the Korea Health Technology R&D Project through

## Abstract

In the backdrop of rapidly changing relapsed/refractory (RR) multiple myeloma (MM) treatment schema that mainly evolves around immunotherapies, it is easy to disregard more traditional drugs. Finding the best partner for pomalidomide, a potent third-generation immunomodulatory drug, is an important agenda we face as a community and cyclophosphamide addition has been used for outcomes augmentation. We carried out this real-world study to identify patients who will show durable response to pomalidomide and those who will benefit from cyclophosphamide addition. A total of 103 patients (57 in pomalidomide-dexamethasone [Pd] group versus 46 in pomalidomide-cyclophosphamide-dexamethasone [PCd]) were studied. They were previously treated with bortezomib (98.1%) or lenalidomide (100%) and previous lines of therapy were median 3 lines. Significantly better overall response rate (ORR) was seen in the PCd (75.6%) than Pd (41.7%) group ($p = 0.001$), but no differences in survival outcomes. Subgroup analysis revealed that high-risk myeloma features, poor response to lenalidomide or bortezomib had superior ORRs when cyclophosphamide was added. Also, long-term responders for pomalidomide were associated with excellent response to previous IMiD treatments. Pomalidomide-based therapy was discontinued in five patients due to intolerance or adverse events, but there was no mortality during treatment. In conclusion, we showed that pomalidomide-based treatment is still relevant and can ensure durable response in RRMM setting, especially for patients who responded well to previous lenalidomide. Addition of cyclophosphamide to Pd is associated with better ORR, and can be positively considered in fit patients with high-risk MM, extramedullary disease, and less-than-satisfactory response to previous lenalidomide treatment.

the Korea Health Industry Development Institute (KHIDI), funded by the Ministry of Health & Welfare, Republic of Korea (grant number: HI14C1277).

**Competing interests:** The authors declare that they have no conflicts of interest to disclose.

## Introduction

Despite recent advances in multiple myeloma (MM) treatment, including monoclonal antibodies [1] and BCMA-targeted immunotherapies [2, 3], treating patients at second relapse and beyond remains complicated. At this point, disease related factors, patient related factors, and effects and toxicity of previous treatments should be taken into consideration. It is also important to highlight that many patients do not have access to newer immunotherapies, thus wisely choosing the optimal treatment sequence among the actually available options deserves equal amount of attention. In this regard, addition of conventional chemotherapy, namely cyclophosphamide, to proteasome inhibitor (PI)-based therapy and/or immunomodulatory drug (IMiD)-based therapy [4–9] has continuously been investigated.

Pomalidomide is a third-generation IMiD with more potent anti-myeloma, anti-inflammatory, and immunomodulatory activities compared to thalidomide and lenalidomide [10, 11]. First attempt to augment the efficacy of pomalidomide-dexamethasone (Pd) regimen by adding cyclophosphamide was undertaken by Baz *et al.*, and they reported significantly improved overall response rate (ORR) in pomalidomide-cyclophosphamide-dexamethasone (PCd) group compared to Pd group (64.7% vs 38.9%, p = 0.0350) without increasing the risk of adverse events (AE) [6]. Encouraged by this study, a phase II AMN001 trial was performed specifically in Asian population, who are often under-represented in multi-national clinical trials [5]. It is particularly important to consider ethnicity and regional bias during cancer treatment because (1) Asian patients manifest different range of hematological and non-hematological AE following chemotherapy [12, 13] and (2) the treatment of hematologic malignancy is costly thus is inevitably influenced by regional health regulation. This trial showed that Pd is well-tolerated in Asian patients but cyclophosphamide addition was not uniformly beneficial.

Resonating such sentiment, we carried out this real-world study to investigate the role of cyclophosphamide addition to Pd in relapsed/refractory (RR) setting. We were especially interested in identifying those who will show durable response to pomalidomide and those who will benefit from cyclophosphamide addition. Korean population was selected, because Korea has a sole public medical insurance system that is mandatory and covers approximately 98% of the overall Korean population and the range of coverage is strictly controlled, thus the MM treatment algorithm is relatively uniform throughout the population [14].

## Materials and methods

### Study design and subjects

This was a single-center retrospective, longitudinal cohort study of RRMM patients over 18 years treated at Seoul National University Hospital. Hundred-and-three patients who were treated with pomalidomide between February 2015 and April 2020 were included (S1 Fig). Their medical records were reviewed and analyzed for demographics, baseline disease characteristics, factors related to MM treatment, response to MM treatment, adverse events, and survival outcomes. This study was performed according to Declaration of Helsinki guidelines and was approved by the Institutional Review Board of Seoul National University Hospital (IRB number H-1912-035-1086). The informed consent was waived in light of the retrospective nature of the study and the anonymity of the subjects.

### Drug administration

Patients were treated with oral pomalidomide 4 mg on days 1–21 and oral or intravenous dexamethasone 40 mg on days 1, 8, 15, and 22 in a 28-day cycle. Oral cyclophosphamide 400

mg was administered on days 1, 8, and 15 in the PCd group. Per attending physician's choice, cyclophosphamide could be added during Pd treatment. The initial dose of pomalidomide or dexamethasone was reduced according to the patient's tolerance. Pomalidomide was withheld if grade 3 or 4 toxicities occurred. It was started again when the toxicities resolved. Pomalido-mide could be reduced to 1–3 mg and dexamethasone to 10–30 mg based on the patient's cir-cumstances. Low-dose aspirin (100mg) and prophylactic ciprofloxacin was routinely prescribed for prophylaxis, unless contraindicated. Patients in PCd group also received an oral serotonin antagonist on days 1, 8, and 15 due to the moderate emetic risk associated with cyclophosphamide [15].

## Response and toxicity evaluation

ORR was defined as the percentage of patients who achieved a stringent complete response (sCR), complete response (CR), very good partial response (VGPR), or partial response (PR) according to International Myeloma Working Group (IMWG) response criteria [16]. High-risk cytogenetics was defined as the presence of del(17p) and/or translocation t(4;14) and/or translo-cation t(14;16) [17]. In addition, high-risk myeloma was defined as International Staging System Stage 3 and/or the presence of extramedullary disease and/or high risk cytogenetics [18]. PFS was defined as the time from administration of pomalidomide-based therapy to disease progres-sion or death from any cause. Lenalidomide PFS was also defined as the time from administra-tion of lenalidomide to disease progression or death from any cause. PFS for the Pd regimen in Pd→PCd group was defined as the time from administration of Pd to the addition of cyclophos-phamide. Patients with long-term PFS was defined as the patients with the top 15% of PFS. Overall survival (OS) was defined as the time from administration of pomalidomide-based ther-apy to death from any cause. Intention-to treatment analysis was performed by grouping patients based on their initial treatment regimens. The AE were assessed according to the National Cancer Institute Common Terminology Criteria for Adverse Events (version 4.03).

## Statistical analysis

Categorical variables were compared using Pearson's chi-squared tests or Fisher's exact tests, as appropriate. Continuous variables were compared using independent or paired *t*-tests, as appropriate. PFS and OS were estimated using the Kaplan-Meier method. If a patient survived without death or progression, survival was censored at the latest date of follow-up. We used median values to determine cut-off values for continuous variables. Clinical variables with *p*-values <0.05 in the univariate analyses were considered for inclusion in multivariate analyses. Cox proportional hazard models were used for the multivariate analyses. All statistical tests were two-sided, and significance was defined as *p*-value <0.05. All analyses were performed using IBM SPSS version 22.0 software (IBM, Armonk, NY, USA).

## Results

### Patient characteristics

Among the 103 patients enrolled, there were 57 in Pd group versus 46 in PCd group. Among the 46 patients in PCd group, 29 received upfront PCd, while in 17 patients, cyclophosphamide was added after median of 6 cycles of Pd (S1 Fig). The median follow-up period was 14.4 months (range, 0.1–51.4 months) and a median of 5 cycles (range 1–30) of pomalidomide-based therapy was delivered. Fifty-three patients (51.5%) tolerated 4 mg pomalidomide until the last dose. In remaining 50 patients, pomalidomide dose was reduced to 3 mg (33 patients), 2 mg (16 patients), or 1 mg (1 patient) due to intolerance or AE.

Baseline characteristics are presented in Table 1. The median age was lower in Pd group (66 years) compared to PCd (71 years, $p = 0.015$). Previous lines of therapy before pomalidomide were median 3 lines (range, 1–11 lines; Table 1). Indicative of Korean medical system, 98.1% of the patients were previously treated with bortezomib and all patients had been exposed to lenalidomide. About half of the study population previously underwent autologous stem cell transplantation (autoSCT).

## Response to treatment

The ORR for all patients was 58.1% (54/93 patients); 3.2% (3/93), 6.5% (6/93), and 48.4% (45/93) patients achieved sCR/CR, VGPR and PR, respectively. PCd group showed a significantly better ORR than Pd group (75.6% vs 41.7%. respectively, $p = 0.001$; Table 2). The subgroup analysis revealed that younger patients (≤68 years), those with a better ECOG performance status (0 or 1), those who did not undergo autoSCT, and those with poor response to lenalido-mide and bortezomib benefitted from cyclophosphamide addition. Also, presence of extreme-dullary disease ($p<0.001$) and high-risk myeloma ($p = 0.003$) favored cyclophosphamide use (Table 2).

## Survival outcomes

The 2-year PFS rates for all patients was 30.6±5.7%. The median PFS was 13.5 months (95% confidence interval [CI], 9.9–17.0 months; Fig 1A). The multivariate analyses (Table 3) revealed that patients with lower Revised-International Staging System (R-ISS) stage and better response to pomalidomide-based therapy had longer overall PFS. More specifically, as shown in S2 Fig, patients achieving PR or better response with pomalidomide-based therapy showed better PFS.

The 2-year OS rates for all patients was 51.4±5.8%. The median OS 25.0 months (95% CI, 17.1–32.8 months; Fig 1B). The multivariate analyses (Table 3) showed high risk cytogenetics and response to pomalidomide were prognostic factors for overall OS (S2 Fig).

Addition of cyclophosphamide did not significantly alter the survival outcomes (Fig 1C–1F). Although the patients who received cyclophosphamide later on (i.e. Pd→PCd group) showed best PFS and OS, the difference did not reach statistical significance. Subgroup analy-ses showed that patients with short lenalidomide PFS duration (<26 months) were likely to benefit from cyclophosphamide addition ($p = 0.048$, S1 Table).

## Response and survival outcomes in Pd→PCd group

We further analyzed patient characteristics, response, and survival in Pd→PCd group (17 patients). All patients in this group received additional cyclophosphamide due to increased M protein before progressive disease. ORR was unchanged in most patients (58.8%) after the addition of cyclophosphamide (VGPR→VGPR for 2/17 patients, PR→PR for 3/17 patients, and SD→SD for 5/17 patients; S2 Table). However, PFS was significantly prolonged after the addition of cyclophosphamide compared with the Pd-only regimen (median 4.0 months for Pd vs. 10.0 months for PCd; S2 Table).

## Intention-to-treatment analysis

We performed intention-to-treatment analysis according to patients' initial treatment regi-mens (Pd group: Pd or Pd→PCd regimens vs. PCd group). PCd group showed significantly better ORR than Pd group (78.6% vs. 49.2%. respectively, $p = 0.009$; S3 Table). Subgroup analy-sis showed that patients with better ECOG performance status (0 or 1), extramedullary disease,

**Table 1. Baseline characteristics of patients.**

| Patient characteristics | All patients (N = 103) | Pd (N = 57) | PCd (N = 46) | p |
|---|---|---|---|---|
| **Median age, years (range)** | 68 (44–85) | 66 (44–82) | 71 (45–85) | 0.015 |
| **Sex, n (%)** | | | | 0.009 |
| Male | 57 (50.4) | 25 (43.9) | 32 (69.6) | |
| Female | 46 (40.7) | 32 (56.1) | 14 (30.4) | |
| **ECOG** | | | | 0.973 |
| 0 | 10 (9.7) | 5 (8.8) | 5 (10.9) | |
| 1 | 81 (78.6) | 45 (78.9) | 36 (78.3) | |
| 2 | 10 (9.7) | 6 (10.5) | 4 (8.7) | |
| 3 | 2 (1.9) | 1 (1.8) | 1 (2.2) | |
| **Extramedullary disease** | | | | 0.425 |
| Presence | 21 (20.4) | 10 (17.5) | 11 (23.9) | |
| Absence | 82 (79.6) | 47 (82.5) | 35 (76.1) | |
| **ISS stage** | | | | 0.429 |
| 1 | 23 (22.3) | 10 (17.5) | 13 (28.3) | |
| 2 | 36 (35.0) | 21 (36.8) | 15 (32.6) | |
| 3 | 37 (35.9) | 22 (38.6) | 15 (32.6) | |
| Unknown | 7 (6.8) | 4 (7.0) | 3 (6.5) | |
| **R-ISS stage** | | | | 0.905 |
| 1 | 10 (9.7) | 5 (8.8) | 5 (10.9) | |
| 2 | 43 (41.7) | 24 (42.1) | 19 (41.3) | |
| 3 | 17 (16.5) | 10 (17.5) | 7 (15.2) | |
| Unknown | 33 (32.0) | 18 (31.6) | 15 (32.6) | |
| **Type of light chains** | | | | 0.356 |
| Kappa | 51 (49.5) | 26 (45.6) | 25 (54.3) | |
| Lambda | 45 (43.7) | 28 (49.1) | 17 (37.0) | |
| Non-secretory | 1 (1.0) | 1 (1.8) | 0 | |
| Unknown | 6 (5.8) | 2 (3.5) | 4 (8.7) | |
| **Isotype of M-protein** | | | | 0.203 |
| IgG / IgA | 51(49.5)/17(16.5) | 29 (50.9)/10 (17.5) | 22 (47.8)/7 (15.2) | |
| IgD / light chain | 7 (6.8)/16 (15.5) | 6 (10.5)/8 (14.0) | 1 (2.2)/8 (17.4) | |
| Non-secretory/Unknown | 1 (1.0)/11 (10.7) | 1 (1.8)/3 (5.3) | 0/8 (17.4) | |
| **Cytogenetics** | | | | 0.204 |
| High risk | 24 (23.3) | 17 (29.8) | 7 (15.2) | |
| Standard risk | 49 (47.6) | 24 (42.1) | 25 (54.3) | |
| Unknown | 30 (29.1) | 16 (28.1) | 14 (30.4) | |
| **Months from diagnosis to pomalidomide, median (range)** | 49 (2–182) | 55 (2–182) | 38 (6–134) | 0.008 |
| **Previous lines of therapy, median (range)** | 3 (1–11) | 3 (1–11) | 2 (2–6) | 0.017 |
| **Previous treatment** | | | | |
| Bortezomib-exposure | 101 (98.1) | 55 (96.5) | 46 (100) | 0.200 |
| Thalidomide-exposure | 48 (46.6) | 32 (56.1) | 16 (34.8) | 0.031 |
| Lenalidomide-exposure | 103 (100) | 57 (100) | 46 (100) | NA |
| Daratumumab-exposure | 4 (3.9) | 1 (1.8) | 3 (6.5) | 0.322 |
| Carfilzomib-exposure | 15 (14.6) | 6 (10.5) | 9 (19.6) | 0.196 |
| Bendamustine-exposure | 2 (1.9) | 2 (3.5) | 0 | 0.501 |
| Previous autoSCT | 46 (44.7) | 32 (56.1) | 14 (30.4) | 0.009 |

Abbreviations: Pd = pomalidomide+dexamethasone; PCd = pomalidomide+cyclophophsamide+dexamethasone; ECOG = Eastern Cooperative Oncology Group performance status; ISS = International Staging System; R-ISS = Revised International Staging System; NA = not applicable; autoSCT = autologous stem cell transplantation.

**Table 2. The Overall Response Rates (ORR) and predictive factors for ORR.**

| Variables (n, %) | | Pd (N = 57) | PCd (N = 46) | p |
|---|---|---|---|---|
| **Response rates** | ORR | 20/48 (41.7) | 34/45 (75.6) | 0.001 |
| | sCR or CR | 1/48 (2.1) | 2/45 (4.4) | 0.609 |
| | VGPR | 2/48 (4.2) | 4/45 (8.9) | 0.425 |
| | PR | 17/48 (35.4) | 28/45 (62.2) | 0.010 |
| | SD | 25/48 (52.1) | 11/45 (24.4) | 0.006 |
| | PD | 3/48 (6.3) | 0 | 0.243 |
| **Age, years** | >68 | 9/17 (52.9) | 19/26 (73.1) | 0.176 |
| | ≤68 | 11/31 (35.5) | 15/19 (78.9) | 0.004 |
| **ECOG** | 0, 1 | 16/41 (39.0) | 30/40 (75.0) | 0.001 |
| | >2 | 4/7 (57.1) | 4/5 (80.0) | 0.576 |
| **Extramedullary disease** | Presence | 1/8 (12.5) | 10/10 (100) | <0.001 |
| | Absence | 19/40 (47.5) | 24/35 (68.6) | 0.066 |
| **R-ISS stage** | 1 | 1/4 (25.0) | 4/5 (80.0) | 0.206 |
| | 2 | 12/22 (54.5) | 14/19 (73.7) | 0.205 |
| | 3 | 3/9 (33.3) | 5/7 (71.4) | 0.315 |
| **High risk myeloma [18]** | High-risk | 11/31 (35.5) | 20/27 (74.1) | 0.003 |
| | None | 9/17 (52.9) | 14/18 (77.8) | 0.164 |
| **Cytogenetics** | High | 5/13 (38.5) | 7/7 (100) | 0.015 |
| | Standard | 7/21 (33.3) | 18/25 (72.0) | 0.009 |
| **Time from diagnosis to pom** | >49 months | 14/29 (48.3) | 12/15 (80.0) | 0.057 |
| | ≤49 months | 6/19 (31.6) | 22/30 (73.3) | 0.004 |
| **Previous treatment lines** | ≥4 | 9/23 (39.1) | 10/14 (71.4) | 0.091 |
| | <4 | 11/25 (44.0) | 24/31 (77.4) | 0.010 |
| **Previous autoSCT** | Done | 11/27 (40.7) | 9/14 (64.3) | 0.153 |
| | Not done | 9/21 (42.9) | 25/31 (80.6) | 0.005 |
| **Previous thalidomide response** | CR/VGPR | 0/2 (0) | 6/7 (85.7) | 0.083 |
| | PR-PD | 6/12 (50.0) | 9/13 (69.2) | 0.428 |
| **Previous lenalidomide response** | CR/VGPR | 4/8 (50.0) | 2/4 (50.0) | 1.000 |
| | PR-PD | 15/39 (38.5) | 32/41 (78.0) | 0.001 |
| **Lenalidomide PFS*** | ≥26months | 4/7 (57.1) | 3/4 (75.0) | 1.000 |
| | <26months | 15/40 (37.5) | 28/36 (77.8) | <0.001 |
| **Previous bortezomib response** | CR/VGPR | 8/16 (50.0) | 13/17 (76.5) | 0.157 |
| | PR-PD | 10/29 (34.5) | 21/28 (75.0) | 0.002 |

*Cut-off of 26 months was used because this was the upper 15% lenalidomide PFS.

Abbreviations: Pd = pomalidomide+dexamethasone; PCd = pomalidomide+cyclophophsamide+dexamethasone; ORR = overall response rate; sCR = stringent CR;
CR = complete response; VGPR = very good partial response; PR = partial response; SD = stable disease; PD = progressive disease; ECOG = Eastern Cooperative
Oncology Group performance status; R-ISS = Revised International Staging System; Pom = pomalidomide; autoSCT = autologous stem cell transplantation;
PFS = progression free survival.

high-risk myeloma, previous treatment lines < 4, or poor response to lenalidomide and borte-zomib benefitted from additional cyclophosphamide (S3 Table). In survival analysis, Pd group showed better OS than PCd group, but PFS was similar between groups (median OS: 27.8 months for Pd group vs. 14.9 months for PCd group, p = 0.040; median PFS: 13.3 months for Pd group vs. 14.0 months for PCd group, p = 0.932; S4 Table).

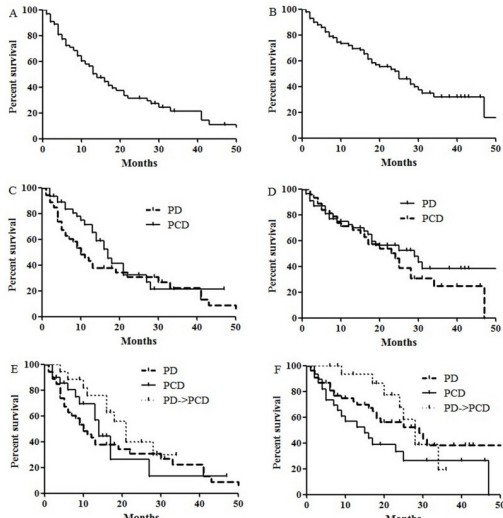

**Fig 1.** (A) Progression-free survival (PFS) and (B) overall survival (OS) of all patients (2-year PFS: 30.6±5.7%, 2-year OS: 51.4±5.8%). (C) Comparison of PFS and (D) OS according to the addition of cyclophosphamide (2-year PFS: 29.7 ±7.4 for Pd vs. 31.5±9.0% for PCd, $p$ = 0.162; 2-year OS: 55.9±7.7% for Pd vs. 46.3±8.6% for PCd, $p$ = 0.358). (E) Comparison of PFS and (F) OS among Pd, PCd and Pd→PCd (2-year PFS: 29.7±7.4% for Pd vs. 26.1±11.9% for PCd vs. 39.4±13.1% for Pd→PCd, $p$ = 0.256; 2-year OS: 55.9±7.7% for Pd vs. 33.1±10.0% for PCd vs. 77.5±11.6% for Pd→PCd, $p$ = 0.111). Abbreviations: Pd = pomalidomide+dexamethasone; PCd = pomalidomide+cyclophophsamide +dexamethasone.

## Prognostic factors for pomalidomide response

In attempt to identify patients who will benefit from pomalidomide-based therapy, we divided the patients into according to pomalidomide PFS regardless of cyclophosphamide use (Table 4). Long-term responders were defined as those with upper 15% PFS ($N$ = 16). For these long-term responders, the median PFS was 32 months (range 25–59 months) in comparison to 5.8 months (range 0–22 months) in all the rest. The long-term responders responded well to previous IMiD treatments: they were associated with better response to previous thalidomide and longer lenalidomide PFS.

## Adverse events

Overall, the most common AE was neutropenia (≥grade 3, 56.7%) followed by a pneumonia (46.6%), thrombocytopenia (≥grade 3, 30.1%), and anemia (≥grade 3, 24.3%) (Table 5). Pomalidomide-based therapy was permanently discontinued for 5 patients, 1 for dyspnea and 4 for intolerance, however there was no mortality during treatment. Addition of cyclophosphamide did not lead to more frequent or severe AE.

## Discussion

The importance of our study lies in that (1) based on real-world experience, we showed that pomalidomide-based treatment is still relevant in this immunotherapy-driven era and can procure durable response in selected group of patients; (2) although cyclophosphamide addition to Pd incurs improved ORR, the results are translated in to prolonged survival thus should be reserved for fit patients with high risk myeloma features; and (3) response to previous lenalidomide treatment can provide guidance to choosing pomalidomide-based therapy and cyclophosphamide addition.

**Table 3. Progression free survival and overall survival in all patients.**

| Variables | | Univariate | | Multivariate | | Univariate | | Multivariate | |
|---|---|---|---|---|---|---|---|---|---|
| | | Median PFS (95% CI) | p | HR (95% CI) | p | Median OS (95% CI) | p | HR (95% CI) | p |
| **Age, years** | >68 | 14.5 (10.5–18.4) | 0.537 | | | 25.0 (18.4–31.5) | 0.282 | | |
| | ≤68 | 12.3 (6.7–17.9) | | | | 25.0 (11.5–38.5) | | | |
| **ECOG** | 0, 1 | 14.0 (10.8–17.2) | 0.657 | | | 25.0 (17.2–32.8) | 0.815 | | |
| | >2 | 10.6 (9.8–11.4) | | | | 28.6 (8.5–48.8) | | | |
| **Extramedullary disease** | Presence | 13.3 (9.2–17.3) | 0.276 | | | 19.8 (13.3–26.2) | 0.048 | 1.628 (0.645–4.112) | 0.302 |
| | Absence | 27.1 (0.1–54.3) | | | | 47.3 (–) | | 1 | |
| **R-ISS stage** | 1 | 21.0 (0.1–55.3) | <0.001 | 1 | | 25.0 (1.3–48.6) | 0.222 | | |
| | 2 | 18.2 (11.1–25.3) | | 2.191 (0.746–6.433) | 0.153 | 23.1 (15.5–30.8) | | | |
| | 3 | 6.1 (3.6–8.7) | | 6.777 (1.966–23.357) | 0.002 | 13.0 (4.7–21.3) | | | |
| **High risk myeloma [18]** | High-risk | 13.5 (8.5–18.4) | 0.961 | | | 19.8 (11.4–28.1) | 0.320 | | |
| | None | 14.0 (9.0–19.0) | | | | 25.3 (20.7–29.8) | | | |
| **Cytogenetics** | Poor | 9.3 (3.6–14.9) | 0.103 | | | 13.3 (3.4–23.2) | 0.014 | 2.158 (1.005–4.633) | 0.048 |
| | Standard | 13.5 (7.8–19.2) | | | | 25.0 (20.1–29.8) | | 1 | |
| **Cyclophophamide** | Added | 16.6 (14.8–18.4) | 0.162 | | | 23.6 (14.7–32.5) | 0.358 | | |
| | Not added | 9.8 (6.1–13.5) | | | | 28.6 (13.9–43.4) | | | |
| **Dx to pomalidomide** | >49months | 14.0 (3.3–24.7) | 0.481 | | | 27.8 (22.9–32.8) | 0.313 | | |
| | ≤49months | 13.3 (9.2–17.3) | | | | 18.8 (10.9–26.7) | | | |
| **Previous treatment lines** | ≥4 | 14.0 (7.6–20.4) | 0.517 | | | 25.3 (17.0–33.5) | 0.717 | | |
| | <4 | 13.5 (7.0–20.0) | | | | 23.1 (11.4–45.0) | | | |
| **Previous autoSCT** | Done | 13.3 (7.0–19.5) | 0.621 | | | 25.3 (16.7–33.8) | 0.268 | | |
| | Not done | 14.5 (9.9–19.0) | | | | 23.1 (12.5–33.8) | | | |
| **Previous thalidomide response** | CR/VGPR | 16.6 (0.1–40.6) | 0.075 | | | 47.3 (–) | 0.087 | | |
| | PR-PD | 13.3 (9.8–16.7) | | | | 18.8 (0.1–38.5) | | | |
| **Previous lenalidomide response** | CR/VGPR | 16.6 (4.1–29.1) | 0.920 | | | 17.7 (4.4–31.1) | 0.882 | | |
| | PR-PD | 13.5 (10.2–16.7) | | | | 25.0 (18.5–31.4) | | | |
| **Previous bortezomib response** | CR/VGPR | 14.5 (9.7–19.2) | 0.410 | | | 23.6 (12.6–34.5) | 0.581 | | |
| | PR-PD | 13.3 (6.8–19.7) | | | | 27.5 (16.1–38.9) | | | |
| **Pomalidomide response** | sCR-PR | 18.2 (8.2–28.2) | <0.001 | 1 | | 23.1 (14.3–32.0) | 0.033 | 1 | 0.008 |
| | SD/PD | 5.5 (1.3–9.8) | | 5.540 (2.600–11.804) | <0.001 | Not reached | | 2.938 (1.325–6.518) | |

Abbreviations: PFS = progression free survival; OS = overall survival; HR = hazard ratio; CI = confidence interval; ECOG = Eastern Cooperative Oncology Group performance status; R-ISS = Revised International Staging System; Dx = diagnosis; autoSCT = autologous stem cell transplantation; sCR = stringent complete response; CR = complete response; VGPR = very good partial response; PR = partial response; SD = stable disease; PD = progressive disease.

The conflicting results from previous reports (Table 6) has prompted us to conduct this real-life study. As an alkylating agent, cyclophosphamide has shown excellent response when combined with Pd with ORR ranging from 65–85% and median PFS of 7–34 months [6–8, 19, 20]. However, these results were primarily from Western population, and recent phase II clinical trial carried out in exclusively Asian patients did not exactly replicate previous benefits of cyclophosphamide addition [5]. In fact, the investigators reported lower ORR in the PCd group (43.6%) compared to Pd group (56.3%) and no significant differences in survival outcomes. In our cohort of patients, cyclophosphamide addition led to improved ORR but no

**Table 4. The comparison between patients with long duration of response to pomalidomide (upper 15% of progression free survival) versus others.**

| Variables | | Long-term responders (N = 16) | Others (N = 87) | p |
|---|---|---|---|---|
| Age, years | | 64 (44–85) | 68 (45–82) | 0.105 |
| R-ISS stage | 1 | 3/9 (33.3) | 7/61 (11.5) | 0.075 |
| | 2 | 6/9 (66.7) | 37/61 (60.7) | |
| | 3 | 0/9 (0) | 17/61 (27.9) | |
| High risk myeloma [18] | High-risk | 12/16 (75.0) | 56/87 (64.4) | 0.568 |
| | None | 4/16 (25.0) | 31/87 (35.6) | |
| Cytogenetics | High | 0/9 (0) | 24/64 (37.5) | 0.025 |
| | Standard | 9/9 (100) | 40/64 (62.5) | |
| Dx to pomalidomide | >49 months | 11/16 (68.8) | 38/87 (43.7) | 0.065 |
| | ≤49 months | 5/16 (31.3) | 49/87 (56.3) | |
| Previous treatment lines | ≥4 | 7/16 (43.8) | 33/87 (37.9) | 0.661 |
| | <4 | 9/16 (56.3) | 54/87 (62.1) | |
| Previous autoSCT | Done | 9/16 (56.3) | 37/87 (42.5) | 0.310 |
| | Not done | 7/16 (43.8) | 50/87 (57.5) | |
| Previous thalidomide response | CR/VGPR | 4/5 (80.0) | 7/34 (20.6) | 0.017 |
| | PR-PD | 1/5 (20.0) | 27/34 (79.4) | |
| Previous lenalidomide response | CR/VGPR | 1/16 (6.3) | 12/85 (14.1) | 0.686 |
| | PR-PD | 15/16 (93.8) | 73/85 (85.9) | |
| Lenalidomide PFS* | ≥26 months | 5/15 (33.3) | 9/82 (11.0) | 0.023 |
| | < 26 months | 10/15 (66.7) | 73/82 (89.0) | |
| Previous bortezomib response | CR/VGPR | 5/16 (31.3) | 29/83 (34.9) | 0.776 |
| | PR-PD | 11/16 (68.8) | 54/83 (65.1) | |
| Cyclophosphamide | Added | 7/16 (43.8) | 39/87 (44.8) | 0.936 |
| | Not added | 9/16 (56.3) | 48/87 (55.2) | |

*Cut-off of 26 months was used because this was the upper 15% lenalidomide PFS.

Abbreviations: R-ISS = Revised International Staging System; autoSCT = autologous stem cell transplantation; PFS = progression free survival; CR = complete response; VGPR = very good partial response; PR = partial response; PD = progressive disease.

differences in PFS or OS, and these results were very similar to Baz *et al.*'s phase II trial results [6]. We do not at this point have a readily available answer for such discrepancy, but we believe our study highlights the importance of real-world data outside of clinical trials setting, albeit being retrospective.

It is also noteworthy that for all patients, the ORR was 58.1% and the median pomalidomide PFS was 13.5 months, which is generally superior compared to previous reports [20–22]. In the backdrop of rapidly changing RRMM treatment schema [23], it is easy to disregard more traditional drugs. However, not all patients have access to emerging immunotherapies including chimeric antigen receptor (CAR) T-cell therapy [24], not to mention the socioeconomic burden that ensues these novel therapeutic options. Effectively triaging patients who can benefit from more conventional treatment is also a challenge that physicians should undertake. Through our study, we identified that previous lenalidomide response is associated with pomalidomide response (i.e. patients who enjoyed durable response with lenalidomide also showed long-term response to pomalidomide). Our result is supported by Kastritis *et al.*, who introduced the concept of "IMiD-sensitive" disease and showed that prior duration of lenalidomide therapy (≥12 months) was associated with longer Pd PFS [25].

**Table 5. Adverse events.**

| Adverse events, n (%) | All patients (N = 103) | Pd (N = 57) | PCd (N = 46) | p |
|---|---|---|---|---|
| Neutropenia (≥gr 3) | 47/103 (56.7) | 25/57 (43.9) | 22/46 (47.8) | 0.613 |
| Anemia (≥gr 3) | 25/103 (24.3) | 14/57 (24.6) | 11/46 (23.9) | 0.989 |
| Thrombocytopenia (≥gr 3) | 31/103 (30.1) | 20/57 (35.1) | 11/46 (23.9) | 0.219 |
| Neutropenic fever | 24/103 (23.3) | 11/57 (19.3) | 13/46 (28.3) | 0.285 |
| Pneumonia | 48/103 (46.6) | 23/57 (40.4) | 25/46 (54.3) | 0.157 |
| Sepsis | 7/103 (6.8) | 3/57 (5.3) | 4/46 (8.7) | 0.697 |
| Kidney injury | 8/103 (7.8) | 6/57 (10.5) | 2/46 (4.3) | 0.293 |
| PPN (≥gr 3) | 1/103 (1.0) | 1/57 (1.8) | 0/46 (0) | 1.000 |
| Peripheral edema (≥gr 3) | 1/103 (1.0) | 1/57 (1.8) | 0/46 (0) | 1.000 |
| Nausea/Vomiting | 9/103 (8.7) | 6/57 (10.5) | 3/46 (6.5) | 0.728 |
| Constipation | 15/103 (14.6) | 9/57 (15.8) | 6/46 (13.0) | 0.694 |
| Diarrhea | 10/103 (9.7) | 6/57 (10.5) | 4/46 (8.7) | 1.000 |

Abbreviations: Pd = pomalidomide+dexamethasone; PCd = pomalidomide+cyclophophsamide+dexamethasone; Gr = grad; PPN, peripheral neuropathy.

One important aspect of our study is that some patients treated with Pd received additional cyclophosphamide (Pd→PCd group). In these patients, the addition of cyclophosphamide before progressive disease tended to prolong the treatment period but did not achieve a

**Table 6. The comparison with previous studies.**

| | Current | AMN001 [5] | IFM2009 [7] | UK series [19] | MM003 [20] |
|---|---|---|---|---|---|
| **Study setting** | Retrospective | Phase II | Phase II | Retrospective | Randomidzed, phase III |
| **Number of patients** | 103 (Pd = 57/PCd = 46) | 136 (Pd = 97/PCd = 39) | 100 | 85 | 302 |
| **Age of all patients (median, range), years** | 68 (44–85) | 66 | 62 (39–70) | 66 (40–89) | 64 (35–84) |
| **Previous exposure to bortezomib** | 101/103 (98.1%) | 135/136 (99.3%) | 100/100 (100%) | 84/85 (98.8%) | 302/302 (100%) |
| **Previous exposure to lenalidomide** | 103/103 (100%) | 136/136 (100%) | 100/100 (100%) | 85/85 (100%) | 302/302 (100%) |
| **Cytogenetic high risk (%)** | 24/73 (32.9%) | 27/44 (61.4%) | 12% | 29/45 (64.4%) | Not available |
| **Treatment** | Pd, PCd | Pd, PCd | PCd | Pd | Pd |
| **Diagnosis to pomalidomide** | 4 years | NA | 3.6 years | 5 years | 5.3 years |
| **Pomalidomide cycles, median (range)** | 4 (2–12) | 7 | (4) | 4 | |
| **Overall response rate, n (%)** | 54/93 (58.1%) | 57/110 (51.8%) | 82/97 (84.5%) | 37/70 (52.9%) | 95/302 (31%) |
| CR | 3/93 (3.2%) | 5/110 (45.5%) | 1/97 (10.3%) | 0/70 | 3/302 (1.0%) |
| VGPR | 6/93 (6.5%) | 13/110 (11.8%) | 32/97 (33.0%) | 4/70 (5.7%) | 14/302 (4.6%) |
| PR | 45/93 (48.4%) | 39/110 (35.5%) | 49/97 (50.5%) | 33/70 (47.1%) | 78/302 (25.8%) |
| **PFS, months (median)** | 13 | 9 | 12 months: 84.1% | 4.5 | 4.0 |
| Pd | 10 | 9 | | | |
| PCd | 17 | 10.8 | 34.2 | | |
| **OS, months (median)*** | 25 | 16.3 | 12 months: 98% | 9.7 | 12.7 |
| Pd | 29 | 15.2 | | | |
| PCd | 24 | 16.3 | NR | | |

*Overall survival defined as time from pomalidomide administration to last follow-up or death.

Abbreviations: CR = complete response; VGPR = very good partial response; PR = partial response; PFS = progression free survival; Pd = pomalidomide +dexamethasone; PCd = pomalidomide+cyclophophsamide+dexamethasone; OS = overall survival.

significantly improved response (S2 Table). However, we found that cyclophosphamide may delay disease progression in patients whose disease is gradually worsening.

Contrary to popular belief that Asian patients are more susceptible to chemotherapy related AE, the AE observed in our group was comparable to previous Western studies [20, 21]. One major difference is the higher rate of pneumonia in our patients. Five (4.9%) patients discontinued pomalidomide in our study, and this incidence rate is also similar to that of previous studies [20, 21].

As shown in S3 Fig, after pomalidomide-based treatment, most patients were treated with carfilzomib-based therapy (23 patients). Ten patients, 6 patients, and 5 patients were treated with bendamustine-based, DCEP (dexamethasone + cyclophosphamide + etoposide + cisplatin), or daratumumab-based therapy, respectively. Six patients underwent other treatments, including melphalan-based (2), thalidomide-based (2), bortezomib-based (1), or cyclophosphamide-based (1) chemotherapy. There were differences in PFS based on the subsequent treatment received (median PFS: 159 days for carfilzomib-based vs. 29 days for daratumumab-based vs. 28 days for bendamustine-based; 186 days for DCEP vs. 34 days for other therapy, $p$ = 0.022). These benefits are probably due to treatment agent-associated differences in resistance mechanisms of MM cells [26]. However, the results should be interpreted with caution because the sample size was small.

The limitations of this study stem from its retrospective nature. First, our study included a small number of patients and had uneven distributions of characteristics between groups, allowing the possibility that bias could influence our results. Thus, studies employing more rigorous designs with larger numbers of patients are needed to confirm our results. Second, there is the innate selection bias as patients were subjected to treatment according to attending physician's choice. Third, evaluation of adverse events was limited because only documented reports could be analyzed. Even so, our findings provide further understandings for physicians to infer decision-making nuances regarding appropriate and realistic RRMM treatment sequence.

## Conclusions

In conclusion, pomalidomide-based therapy can ensure durable response in RRMM setting, especially for patients who responded well to previous lenalidomide. Addition of cyclophosphamide to Pd is associated with better ORR, and can be positively considered in fit patients with high risk MM, extramedullary disease, and less-than-satisfactory response to previous lenalidomide treatment. Our next agenda regards on identifying the best partner for pomalidomide.

## Supporting information

**S1 Table. PFS and OS according to cyclophosphamide addition.**
(DOCX)

**S2 Table. Baseline characteristics, response and survival outcomes of patients in Pd→PCd group.**
(DOCX)

**S3 Table. The Overall Response Rates (ORR) and predictive factors for ORR (Intention-to-treatment analysis).**
(DOCX)

**S4 Table. Progression free survival and overall survival in all patients (Intention-to-treatment analysis).**
(DOCX)

**S1 Fig. CONSORT diagram.**
(DOCX)

**S2 Fig. PFS and OS according to pomalidomide response.**
(DOCX)

**S3 Fig. Subsequent treatment after pomalidomide-based therapy.**
(DOCX)

## Author Contributions

**Conceptualization:** Ja Min Byun, Sung-Soo Yoon, Youngil Koh.

**Data curation:** Hyunkyung Park, Ja Min Byun, Youngil Koh, Sock-Won Yoon, Dong-Yeop Shin, Junshik Hong, Inho Kim.

**Formal analysis:** Hyunkyung Park, Ja Min Byun.

**Funding acquisition:** Sung-Soo Yoon.

**Supervision:** Sung-Soo Yoon, Youngil Koh.

**Writing – original draft:** Hyunkyung Park, Ja Min Byun.

**Writing – review & editing:** Dong-Yeop Shin, Junshik Hong, Inho Kim.

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
