## [Decision Letter · Decision Letter 0]

16 Aug 2021

PONE-D-21-18676

Cyclophosphamide addition to pomalidomide/dexamethasone is not necessarily associated with universal benefits in RRMM

PLOS ONE

Dear Dr. Yoon,

Thank you for submitting your manuscript to PLOS ONE. After careful consideration, we feel that it has merit but does not fully meet PLOS ONE’s publication criteria as it currently stands. Therefore, we invite you to submit a revised version of the manuscript that addresses the points raised during the review process.

We look forward to receiving your revised manuscript.

Kind regards,

David Dingli

Academic Editor

PLOS ONE

Journal Requirements:

"This study was supported by a grant of the Korea Health Technology R&D Project through the Korea Health Industry Development Institute (KHIDI), funded by the Ministry of Health & Welfare, Republic of Korea (grant number: HI14C1277)."

"This study was supported by a grant of the Korea Health Technology R&D Project through the Korea Health Industry Development Institute (KHIDI), funded by the Ministry of Health & Welfare, Republic of Korea (grant number: HI14C1277)."

"This study was supported by a grant of the Korea Health Technology R&D Project through the Korea Health Industry Development Institute (KHIDI), funded by the Ministry of Health & Welfare, Republic of Korea (grant number: HI14C1277)."

"This study was supported by a grant of the Korea Health Technology R&D Project through the Korea Health Industry Development Institute (KHIDI), funded by the Ministry of Health & Welfare, Republic of Korea (grant number: HI14C1277)."

Additional Editor Comments (if provided):

The manuscript has been reviewed by two experts in the field. Please see their comments below and the concerns have to be addressed in a satisfactory manner for the manuscript to be accepted for publication.

Thank you for considering PLoS ONE for your work.

Sincerely

David Dingli

Reviewers' comments:

Reviewer's Responses to Questions

**Comments to the Author**

1. Is the manuscript technically sound, and do the data support the conclusions?

Reviewer #1: Yes

Reviewer #2: Partly

2. Has the statistical analysis been performed appropriately and rigorously? 

Reviewer #1: Yes

Reviewer #2: No

3. Have the authors made all data underlying the findings in their manuscript fully available?

Reviewer #1: No

Reviewer #2: Yes

4. Is the manuscript presented in an intelligible fashion and written in standard English?

Reviewer #1: Yes

Reviewer #2: Yes

5. Review Comments to the Author

Reviewer #1: I read with interest the manuscript by Park et al. The authors studied in retrospective study the added value of adding CTX to pomalidomide and dexamethasone for relapsed refractory MM patients. The authors concluded that adding CTX increases response rate but no survival outcome. to note, that majority of response enhancement is at a level of partial response. The authors also found that patients with high-risk cytogenetics and poorer response to prior IMiD increases response to added CTX.

The paper is overall well written and easy to follow.

I have several concerns as outlined below:

1. The Two groups (Pd, PCd) are not balanced with regard to age, number of prior lines of therapy/months from diagnosis to Pomalidomide initiation and prior ASCT. This is a major caveat and the authors should try and explain these differences and highlight it in the text. This is one of the unavoidable limitation of retrospective study, and number of patients (n=103) is too small to partially overcomes these biases.

2. The authors performed MVA to find independent predictors to PFS and OS. However, the used 2-year PFS/OS which is less informative. The authors should have perform time-to-event analysis. Also choosing P-value of 0.05 in the univariate analysis is very restrictive. I am not sure the authors have enough power for MVA, but being under-powered does not justify less than satisfactory analysis.

3. 17 patients (out of 46 patients in the PCd group) started with Pd and CTX was added at a later stage. This is a major bias, since we do not know what CTX would add to the comparative group. The authors should list in detail the results of this subgroup, including their baseline characteristics, response (survival is indeed listed).

Reviewer #2: Review of article D-21-18676

This represents a retrospective review that attempts to compare patients treated with doublet pomalidomide dexamethasone with those treated with triplet cyclophosphamide pomalidomide dexamethasone.

The authors attempt to compare 57 in the doublet group compared with 46 in the triplet group and conclude that response rate was higher in the triplet group and try to identify subsets that benefit with the addition of cyclophosphamide.

Please clarify why patients receiving the triplet received an oral serotonin antagonist. This is found on page 5

The authors will need to reconsider the analysis because the groups they are reporting are not “intent to treat”. Nearly 35% of the patients in the pomalidomide cyclophosphamide dexamethasone group had 17 patients where cyclophosphamide was added after median of 6 cycles of pomalidomide dexamethasone. This distorts the statistical analysis. This was clearly “intent to treat” with a doublet and the addition of cyclophosphamide suggest these patients had an inadequate response to the doublet so this represents clear reporting bias. Moreover, patients would need to survive for 6 months in order to begin cyclophosphamide which will impact their report of progression-free and overall survival making it look better for triplet group since the timing of progression-free survival would have been initiation of the doublet 6 cycles earlier. The report of an overall response rate of 75.6% in the triplet group would also be biased based on what I assume would be a failure to respond to the doublet and the addition of pomalidomide at 6 months.

Therefore, the analysis really needs to be intention to treat and the 17 patients that had cyclophosphamide added after 6 months belong to the doublet group of pomalidomide dexamethasone. I suspect this will have a profound impact on the interpretation of the data. Since the median pomalidomide progression-free survival was 13.5 months and pomalidomide was added to 17 patients after 6 months this will almost certainly decline significantly.

Finally, retrospective trials are notoriously poor in reporting toxicity since it requires documentation by the provider on side effects that may not have been pre specified in the trial such as nausea vomiting diarrhea weight loss etc the discussion should indicate that in a retrospective trial documenting toxicity is not particularly reliable.

6. PLOS authors have the option to publish the peer review history of their article (what does this mean?). If published, this will include your full peer review and any attached files.

Reviewer #1: **Yes: **Eli Muchtar

Reviewer #2: No

---

## [Author Response · Author response to Decision Letter 0]

13 Sep 2021

Thank you for your comments. We attached the file.

---

## [Editor Report · Decision Letter 1]

3 Nov 2021

Cyclophosphamide addition to pomalidomide/dexamethasone is not necessarily associated with universal benefits in RRMM

PONE-D-21-18676R1

Dear Dr. Yoon,

We’re pleased to inform you that your manuscript has been judged scientifically suitable for publication and will be formally accepted for publication once it meets all outstanding technical requirements.

Kind regards,

David Dingli

Academic Editor

PLOS ONE
---

## [Editor Report · Acceptance letter]

10 Nov 2021

PONE-D-21-18676R1 

Cyclophosphamide addition to pomalidomide/dexamethasone is not necessarily associated with universal benefits in RRMM 

Dear Dr. Yoon:

I'm pleased to inform you that your manuscript has been deemed suitable for publication in PLOS ONE. Congratulations! Your manuscript is now with our production department. 

Kind regards, 

on behalf of

Dr. David Dingli 

Academic Editor

PLOS ONE